# Are rapid tests and confirmatory western blot used for cattle and small ruminants TSEs reliable tools for the diagnosis of Chronic Wasting Disease in Europe?

**Maria Mazza**[1]*, **Linh Tran**[2], **Daniela Loprevite**[1], **Maria C. Cavarretta**[1], **Daniela Meloni**[1], **Luana Dell'Atti**[1], **Jørn Våge**[2], **Knut Madslien**[2], **Tram T. Vuong**[2], **Elena Bozzetta**[1‡], **Sylvie L. Benestad**[2‡]

**1** European Reference Laboratory for Transmissible Spongiform Encephalopathies - Italian Reference Laboratory for Transmissible Spongiform Encephalopathies, Istituto Zooprofilattico Sperimentale del Piemonte, Liguria e Valle d'Aosta, Turin, Italy, **2** World Organisation for Animal Health (WOAH, founded as OIE) - Reference Laboratory for Chronic Wasting Disease, Norwegian Veterinary Institute, Oslo, Norway

☯ These authors contributed equally to this work.
‡ EB and SLB also contributed equally to this work.
* maria,mazza@izsto.it

**Data Availability Statement:** All relevant data are within the paper and its Supporting information files.

## Abstract

The first case of CWD in Europe was detected in a Norwegian reindeer in 2016, followed later by two CWD cases in Norwegian moose. To prevent the potential spread of CWD to the EU, the European Commission (Regulation EU 2017_1972) implemented a CWD surveillance programme in cervids in the six countries having reindeer and or moose (Estonia, Finland, Latvia, Lithuania, Poland, and Sweden). Each country had to test a minimum of 3000 cervids for CWD using diagnostic rapid tests approved by the EC Regulation. Experimental transmission studies in rodents have demonstrated that the CWD strains found in Norwegian reindeer are different from those found in moose and that these European strains are all different from the North American ones. Data on the performances of authorised rapid tests are limited for CWD (from North America) and are currently minimal for CWD from Europe, due to the paucity of positive material. The aim of this study was to evaluate the diagnostic performances of three of the so-called "rapid" tests, commercially available and approved for TSE diagnosis in cattle and small ruminants, to detect the CWD strains circulating in Europe. The performances of these three tests were also compared to two different confirmatory western blot methods. Using parallel testing on the same panel of available samples, we evaluated here the analytical sensitivity of these methods for TSE diagnosis of CWD in Norwegian cervids tissues. Our results show that all the methods applied were able to detect the CWD positive samples even if differences in analytical sensitivity were clearly observed. Although this study could not assess the test accuracy, due to the small number of samples available, it is conceivable that the rapid and confirmatory diagnostic systems applied for CWD surveillance in Northern Europe are reliable tools.

**Funding:** This study was partially funded by the Norwegian Ministry of Agriculture (project 12081) and by the Italian Ministry of Health (grant RF-2019-12369570). The funders had no role in study design, data collection and analysis, decision to publish, or preparation of the manuscript.

**Competing interests:** The authors have declared that no competing interests exist.

## Introduction

Chronic Wasting Disease (CWD) is a fatal neurodegenerative disease that affects different cervid species and the most predominant clinical sign is emaciation. CWD belongs to the group of Transmissible Spongiform Encephalopathies (TSEs) or prion diseases, that affects animals and humans, including bovine spongiform encephalopathy (BSE) in cattle, scrapie in sheep and goats, and Creutzfeldt-Jakob disease (CJD) in humans.

The etiological agent of TSEs, called prion, is the misfolded pathogenic form ($PrP^{Sc}$) of the host-encoded cellular prion protein ($PrP^{C}$). Prion accumulation in the central nervous system leads to neurodegeneration and, eventually, to death [1–4].

CWD, like classical Scrapie, is a highly contagious disease under natural conditions and can be efficiently transmitted between cervids through direct and environmental contacts. The disease involves not only the central nervous system but also the lymphoreticular system. Presence of the pathological prion or prion seeding activity has been found in saliva, feces and urine, and the agent can persist in the environment for many years, increasing the risk of exposure also for other animal species [5–9].

CWD was detected for the first time in 1967 in a mule deer (*Odocoileus hemionius*) in Colorado and since then in additional cervid species and the disease has expanded its geographic distribution with currently 30 states in U.S., four provinces in Canada (Saskatchewan 1996, Alberta 2005, Quebec 2018, Manitoba 2021), South Korea, and more recently Norway, Finland and Sweden [10–12] [https://www.usgs.gov/media/images/distribution-chronic-wasting-disease-north-america-0].

The first case of CWD in Europe was identified in 2016 in a free-ranging reindeer (*Rangifer tarandus*) in Norway [11]. A total of 35 CWD cases have been identified in Norway to date: 21 reindeer (*Rangifer tarandus)*, 11 moose (*Alces alces*) and three red deer (*Cervus elaphus*). Three cases of CWD have been reported in Finland since March 2018 and four cases are reported in Sweden since March 2019, all in moose [13, 14].

To prevent the spread of CWD within the EU, and/or to control the disease where it occurs, the European Commission implemented a CWD surveillance program in cervids in the six countries having reindeer and/or moose (Estonia, Finland, Latvia, Lithuania, Poland, and Sweden). This survey was performed in 2018–2020, by using diagnostic rapid tests approved by the Commission Regulation (EU) 2017/1972 [15].

The persistent expansion of CWD in North America and the emergence of the disease in Nordic countries emphasize the need for efficient management options, which are highly dependent on performant diagnostic tools.

The European TSEs Regulation (EC N˚ 999/2001) establishes that each EU Member State shall carry out an annual monitoring programme for TSEs in small ruminants and cattle based on rapid tests, that allow results to be available within 24 hours. International guidelines for validation of diagnostic tests for infectious diseases in animals are described in the OIE Terrestrial Manual, in chapter 1.1.6. (OIE, 2018). It is specified that the tests should be validated for the species in which they will be used. There are three rapid tests that are currently commercially available and approved for the diagnosis of TSE in cattle and small ruminants: TeSeE™ SAP Combi Kit (Bio-Rad), TeSeE™ Sheep/Goat, (Bio-Rad), HerdChek BSE-Scrapie Antigen (Ag) test (IDEXX). They have been extensively validated for cattle, and in a lesser extend in small ruminants. Data on the performances of authorised rapid tests in North America cervids are not exhaustive and there are no direct comparisons of rapid test performances available in cervids affected with CWD [16]. In addition, due to insufficient positive reference samples from European CWD samples, the evaluation of the performances of the different available tests for cervid samples has not been requested to date.

Given the unusual biochemical characteristics of the European CWD isolates and the demonstration by inoculation into rodents that the European CWD strains are not identical to the North American [17–19], information about the diagnostic performances of the above tests are highly needed. In the present study, we evaluated the analytical sensitivity of methods for TSE diagnosis in Norwegian cervids tissues, on two sample sets. Due to the paucity of nervous tissue from CWD positive cases, it was not possible to apply the same diagnostic methods on both sample sets. The first set was analysed with all commercially available ELISA tests TeSeE™ SAP Combi Kit (Bio-Rad), TeSeE™ Sheep/Goats (Bio-Rad), HerdChek BSE-Scrapie Ag test (IDEXX) and HerdChek CWD Ag test (IDEXX). The second set of samples was tested in parallel in two different laboratories using the three commercial rapid tests: TeSeE™ SAP Combi Kit (Bio-Rad), TeSeE™ Sheep/Goats (Bio-Rad) and HerdChek BSE-Scrapie Ag test (IDEXX) and two confirmatory western blot methods, the one commercially available, TeSeE™ Western Blot (Bio-Rad) and one Scrapie Associated Fibrils (SAF) Immunoblot, developed at the Italian TSE Reference Laboratory (IRL).

## Materials and methods

### Animal and tissues

A total of five moose and two reindeer, detected as CWD positive through the Norwegian surveillance programme, were included in this study. Due to the lack of remaining material from the medulla oblongata, each sample was represented from a pool from several regions of the brain, particularly the cerebral cortex. The reindeer analyses were carried out on the brain tissue from two animals and one retropharyngeal lymph node. Negative samples were made from pooled brain tissues from six moose and six reindeer for the set 1 and from 14 moose for set 2 and were included in the analyses as negative controls. Table 1 shows the list of animals included in this study.

For the analyses by rapid tests and confirmatory western blot two different sets of homogenized samples were prepared, as shown in Fig 1.

**Set 1** prepared at the Norwegian Veterinary Institute (NVI), included brain material from two moose, two reindeer and a retropharyngeal lymph node from one of the two reindeer.

The central nervous tissue from each animal was thoroughly chopped and mixed well until the tissue appeared homogeneous, before being distributed into either Bio-Rad or IDEXX grinding tubes according to the producers' recommendations and homogenised using a TeSeE™ Precess 24 homogenizing system. Lymph node tissue was homogenised using an additional single large (6mm diameter) ceramic bead to give 20% (w/v) homogenate. Each sample was diluted in negative brain material as a 2 base logarithm dilutions series from 1:2 to 1:128 dilutions and with the following ELISA tests: TeSeE SAP, TeSeE Sheep/goat, IDEXX HerdChek Bovine conjugate, IDEXX HerdChek SR conjugate, IDEXX HerdChek short protocol, IDEXX CWD. Each sample was analysed in duplicates and an average optical density (OD) value was calculated as recommended by the producers.

**Set 2**, prepared at the Italian TSE Reference Laboratory (IRL), included brain material from five moose. Brain samples were subjected to a pre-homogenisation protocol. A 50% w/v homogenate was made from CWD brain tissues in distilled water. The analytical sensitivity of the tests was performed from a dilution series of the brain diluted with CWD negative cervid brain homogenate as a 2 base logarithm dilutions series down to 1:128. Each dilution was submitted to various cycle of homogenization (the first at low speed, the second at medium speed, and the third at high speed, with a 30-second interval in between) to ensure the preparation was thoroughly mixed. The resulting homogenate was aliquoted into pre-labelled cryotubes or

**Table 1. List of negative and positive natural CWD cases included in the study.**

| Species | CWD status | ID Number | Code | Prp genotype | Sex | Area | Set 1 | Set 2 |
|---|---|---|---|---|---|---|---|---|
| Moose | positive | 16-P138 | Moose A | $KK_{109}$ | Female | Selbu | X | X |
| | positive | 16-P153 | Moose B | $KK_{109}$ | Female | Selbu | | X |
| | positive | 17-CD11399 | Moose C | $KK_{109}$ | Female | Lierne | X | X |
| | positive | 19-CD24854 | Moose D | $QQ_{109}$ | Female | Sigdal | | X |
| | positive | 20-CD3380 | Moose E | $KK_{109}$ | Female | Steinkjer | | X |
| Reindeer | positive | 17-CD2788 | Reindeer A | A/C | Male | Nordfjella | X | |
| | positive | 17-CD20830 | Reindeer B | C/C | Male | Nordfjella | X | |
| | positive | 17-CD20831 lymph node | Reindeer B lymph node | C/C | Male | Nordfjella | X | |
| Moose | negative | 20-CD4385 | //// | n.a. | n.a. | n.a. | | X |
| | negative | 20-CD4384 | //// | n.a. | n.a. | n.a. | | X |
| | negative | 20-CD4380 | //// | n.a. | n.a. | n.a. | | X |
| | negative | 20-CD4379 | //// | n.a. | n.a. | n.a. | | X |
| | negative | 20-CD38 | //// | n.a. | n.a. | n.a. | | X |
| | negative | 20-CD97 | //// | n.a. | n.a. | n.a. | | X |
| | negative | 18-80-55 | //// | n.a. | n.a. | n.a. | X | |
| | negative | 18-80-58 | //// | n.a. | n.a. | n.a. | X | |
| | negative | 18-04-V179 | //// | n.a. | n.a. | n.a. | X | |
| Reindeer | negative | 18-80-43 | //// | n.a. | n.a. | n.a. | X | |
| | negative | 18-80-57 | //// | n.a. | n.a. | n.a. | X | |
| | negative | 18-80-80 | //// | n.a. | n.a. | n.a. | X | |
| | negative | 18-80-78 | //// | n.a. | n.a. | n.a. | X | |

lf: lymph node; n.a.: not available; A and C: *PRNP* alleles according to Güere et al. 2020 [20].

distributed into either Bio-Rad or IDEXX grinding tubes according to the producers' recommendations. The test was performed as per the manufacturer's manual method instructions.

All dilutions of each moose sample were analysed in parallel at the NVI and the Italian Reference Laboratory by the following tests: TeSeE™ SAP Combi Kit (Bio-Rad), TeSeE™ Sheep/

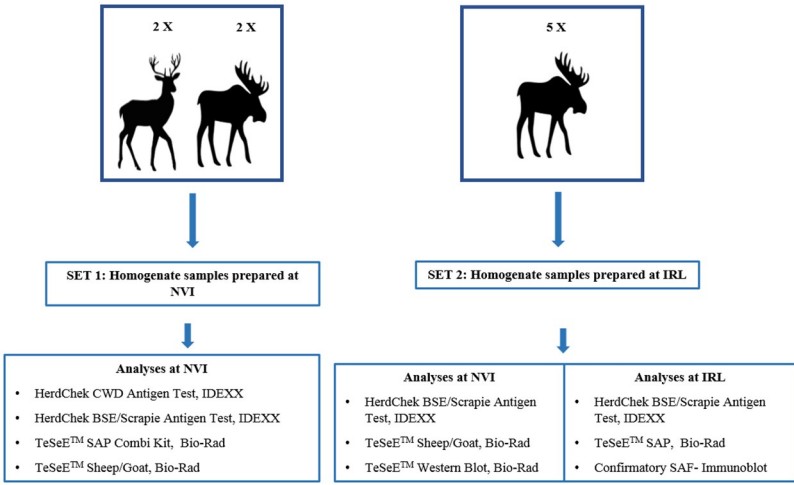

**Fig 1. Overview about the CWD positive animals and tests used in this study.**

Goat (Bio-Rad) and HerdChek BSE-Scrapie Ag test (IDEXX). Each sample was analysed in duplicate or triplicate with the three screening tests and an average of OD value was calculated as recommended by the producers.

In addition, two different confirmatory western Blot methods, TeSeE™ Western Blot kit (Bio-Rad), commercially available, and SAF-Immunoblot were performed at the NVI and IRL, respectively.

**Rapid tests.** TeSeE™ SAP Combi Kit and TeSeE™ Sheep/Goat rapid tests are diagnostic methods based on a homogenate digestion step of PrP$^C$ with proteinase K (PK) to select for PrP$^{Sc}$, which is partially resistant to PK action due to its β-sheet structure and its aggregation formation. In contrast, Herdchek BSE-Scrapie Ag test does not involve any digestion with PK but uses a particular ligand that can capture PrP$^{Sc}$ by a specific conformational recognition of PrP$^{Sc}$ aggregates.

*TeSeE methods.* TeSeE™ SAP Combi and TeSeE™ Sheep/Goat are ELISA sandwich techniques. TeSeE™ SAP was applied at the IRL using a manual protocol while the TeSeE™ Sheep/Goat was carried out at the NVI using robotic system as NSP and EVOLIS delivered by Bio-Rad. The protocol was similar for both methods, but the kits use different reagents in the immunodetection step. Briefly, 250 μl of the homogenate sample were incubated for 10 minutes at 37 ˚C with 250 μl of denaturing solution, buffer A /reagent 1 for TeSeE SAP and TeSeE Sheep /Goat respectively, containing PK. The digestion was stopped by addition of 250 μl of clarifying solution buffer B/reagent 2. PrP$^{Sc}$ was recovered as a pellet after the micro test-tubes were centrifuged at 20000 g for 5 minutes at room temperature. The supernatant was discarded, and the tubes dried. Finally, the pellet was denatured in 25 μl resolving buffer C/ reagent 3 (5 minutes at 100 ˚C) then diluted with 125 μl sample diluent reagent R6 before 100 μl of it were distributed into the ELISA wells. The immunodetection part was performed 30 minutes at 37 ˚C, washes, 100 μl of conjugate solution R7 and incubation 30 minutes at between 2 and 8 ˚C, washes, before 100 μl of the enzymatic revelation solution (R8+R9) was applied for 30 minutes in darkness at room temperature. The revelation process was stopped by adding 100 μl of stop solution (R10) to each well and the absorbance was read at 450nm and 620nm. Samples with an OD lower than the cut-off value are considered to be negative; samples with an OD greater than or equal to the cut-off value are considered to be positive. Calculation of the cut-off value was carried out according to the manufacturer's instructions.

*HerdChek BSE-Scrapie Ag test and HerdChek CWD Ag test.* The test protocol for these two tests is the same with the exceptions of longer incubation times in the immunological detection phase and a different cut-off value. The tests were carried out according to the manufacturer's instructions. Briefly, 120 μl of homogenates was mixed with 30 μl of the working plate diluent solution (D1 and D2), and 100 μl of the mixture were loaded on to the antigen-capture plate and shaken for 45 minutes at room temperature. After washes, the plate was incubated in 100 μl of conditioning buffer (CB) for 10 minutes. Abnormal PrP was detected using 100 μl of the kit conjugated anti-PrP antibody, Conjugate concentrate (CC) (incubation of 45 minutes for HerdChek BSE-Scrapie Ag and 60 minutes for HerdChek CWD Ag), visualised with 100 μl TMB (tetramethylbenzidine) for 15 minutes of incubation in darkness and absorbance read at 450nm and 620nm. Interpretation of sample results is based on absorbance for the sample. Samples with OD less than the cut-off value are considered negative; samples with OD values greater than or equal to the cut-off are classified as positive. Calculation of the cut-off value was carried out according to the manufacturer's instructions.

**Confirmatory western blot methods.** *TeSeE$^{TM}$ western blot, Bio-Rad.* The test was carried out according to the manufacturer's instructions with slight modifications. Five hundred μl reagent A/PK solution were added to 500 μl homogenate before incubation at 37 ˚C for 10

minutes. For the moose isolates, half the PK concentration was used rather than the one specified in the protocol, as this change resulted in a more pronounced PrP$^{Sc}$ signals.

At the end of the incubation, 500 μl buffer B was added and the tubes centrifuged for 7 minutes at 15000 g. The supernatant was discarded, 100 μl Laemmli solution was added to the pellet and left for 5 minutes at room temperature before heating for 5 minutes at 100 ˚C, followed by 15 minutes centrifuging at 15000 g. The supernatant was collected and heated at 100 ˚C for 4 minutes before loading onto MINI PROTEAN TGX Precast Protein Gels (Bio-Rad) for electrophoretic separation, and then transferred onto polyvinylidene fluoride membrane (PVDF) using a Trans-Blot Turbo Midi PVDF Transfer Packs (Bio-Rad). Immunodetection was then performed as described in the kit´s instructions (incubations with primary and secondary antibody of 30 and 20 minutes respectively) with the monoclonal antibody (mAb) Sha31 which recognizes the 145–152 sequence of PrP (YEDRYYRE) before revelation with Enhanced Chemiluminescence (ECL) SuperSignal West Pico Plus substrate (Thermo Fisher Scientific, Invitrogen) and visualization read by a Chemidoc MP (Multiplex fluorescence) Imager (Bio-Rad).

*SAF-Immunoblot*. 10% (w/v) homogenates of brain tissue were prepared in lysis buffer [10% N-lauroylsarcosine diluted in Tris-buffered saline (TBS), pH 7.4] and clarified by centrifugation at 22000 g for 20 minutes at 10 ˚C. 1 ml of each supernatant was digested by PK (40 μg per ml) at 37 ˚C for 1 h. The samples were then centrifuged at 215000 g for 1 hour at 10 ˚C; the pellets were dissolved in 50 μl of Laemmli buffer and 50 μl of distilled water. 10 μl (corresponding to 10 mg of tissue) of this extract were subjected to sodium dodecyl sulfate-polyacrylamide gel electrophoresis by Nu-PAGE Bis-Tris mini-gels (Thermo Fisher Scientific, Invitrogen) and then transferred onto polyvinylidene difluoride membranes using a Trans-Blot Turbo Transfer System (Bio-Rad). PrP$^{Sc}$ immunodetection was performed overnight at 4 ˚C using five different monoclonal antibodies (mAbs) with different epitopes (sheep PrP numbering): SAF84 (aa 167–172) was obtained from Cayman Chemical Co., diluted 1:1000; 6H4 (aa 153–165) from Thermo Fisher Scientific, Prionics, diluted 1:10000; Sha31 (aa148-155) from SpiBio, France, diluted 1:2000; L42 (aa 148–153) from R-Biopharm, diluted 1:1000 and 9A2 (aa 102–104) from Wageningen Bioveterinary research, Lelystad, Netherlands, diluted 1:5000. Immunosignals were revealed with an alkaline phosphatase-conjugated goat antimouse immunoglobulin G (0.1 μg per ml) (Thermo Fisher Scientific, Invitrogen) and immuno-reactivity was visualized by a chemiluminescent reaction with Novex® AP Chemiluminescent Substrate CDP-Star® (ThermoFisher Scientific, Invitrogen). The images of the blots were captured with a gel documentation analysis system (iBright, ThermoFisher Scientific). Samples were classified positive when at least the di-glycosylated band of PrP$^{Sc}$ was present.

## Results

### Rapid test

The results of the diagnostic investigations performed at the NVI on the set 1 are shown in Table 2. The data obtained revealed that prions were detected in all dilutions of the two positive moose samples when analysed by HerdChek BSE-Scrapie Ag test and especially with the HerdChek CWD Ag test. A lower sensitivity (up to 1:32 and 1:64) on these samples was observed with the TeSeE™ Sheep/Goat test, while only undiluted samples were detected with the TeSeE™ SAP Combi kit.

Comparable analytical sensitivity was found for the HerdChek BSE-Scrapie Ag and TeSeE™ Sheep/Goat tests in both moose and reindeer samples, except for the reindeer Reindeer A probably due to low PrP$^{Sc}$ in the brain. TeSeE™ SAP Combi test showed better performance in reindeer than in moose, especially for the analysis of the lymph node sample.

**Table 2. Results obtained from the set 1 of samples by the three ELISA rapid tests at NVI.** The O.D. represents the mean of duplicates values obtained from each dilution sample. The values above the cut-off indicate positive sample and those below the cut-off indicate negative sample.

| ID sample | Dilution | HerdCheck CWD | HerdCheckBSE/Scrapie Ag Test with Bovine conjugate, ultra short protocol | HerdCheckBSE/Scrapie Ag Test with Bovine conjugate, short protocol | HerdCheckBSE/Scrapie Ag Test with small ruminants conjugate | TeSeE™ SAP | TeSeE™ Sheep & Goat |
|---|---|---|---|---|---|---|---|
| Moose A | 1 = 2 | 3, 45 | 3,5 | 3,5 | 3,382 | 0,069 | 2,788 |
| | 1 = 4 | 3,445 | 3,274 | 3,387 | 3,095 | 0,06 | 2,143 |
| | 1 = 8 | 3,247 | 1,659 | 3,18 | 2,675 | 0,057 | 1,159 |
| | 1 = 16 | 2,958 | 1,022 | 2,482 | 1,749 | 0,025 | 0,808 |
| | 1 = 32 | 2,027 | 0,569 | 1,364 | 1,072 | 0,012 | 0,442 |
| | 1 = 64 | 1,148 | 0,431 | 0,967 | 0,62 | 0,012 | 0,182 |
| | 1 = 128 | 1,037 | 0,305 | 0,773 | 0,556 | 0,014 | 0,077 |
| Moose C | 1 = 2 | 3,5 | 3,5 | 3,5 | 3,392 | 0,04 | 2,783 |
| | 1 = 4 | 3,453 | 3,269 | 3,5 | 3,219 | 0,03 | 2,283 |
| | 1 = 8 | 3,366 | 1,85 | 3,27 | 2,632 | 0,02 | 1,695 |
| | 1 = 16 | 2,94 | 1,186 | 2,699 | 1,801 | 0,016 | 0,77 |
| | 1 = 32 | 2,1418 | 0,595 | 2,071 | 1,324 | 0,012 | 0,702 |
| | 1 = 64 | 1,711 | 0,404 | 1,402 | 0,936 | 0,015 | 0,093 |
| | 1 = 128 | 0,899 | 0,187 | 0,815 | 0,483 | 0,009 | 0,043 |
| Reindeer A | 1 = 2 | 1,749 | 0,262 | 1,655 | 1,267 | 0,019 | 0,181 |
| | 1 = 4 | 0,936 | 0,106 | 0,841 | 0,506 | 0,016 | 0,115 |
| | 1 = 8 | 0,487 | 0,059 | 0,456 | 0,318 | 0,014 | 0,05 |
| | 1 = 16 | 0,259 | 0,039 | 0,245 | 0,199 | 0,011 | 0,038 |
| | 1 = 32 | 0,148 | 0,038 | 0,131 | 0,118 | 0,011 | 0,03 |
| | 1 = 64 | 0,089 | 0,04 | 0,08 | 0,069 | 0,008 | 0,03 |
| | 1 = 128 | 0,062 | 0,035 | 0,059 | 0,054 | 0,009 | 0,015 |
| Reindeer B | 1 = 2 | 3,11 | 2,837 | 3,102 | 2,343 | 1,97 | 2,793 |
| | 1 = 4 | 2,536 | 1,825 | 1,743 | 1,383 | 1,081 | 2,651 |
| | 1 = 8 | 1,422 | 0,619 | 0,963 | 0,791 | 0,514 | 1,136 |
| | 1 = 16 | 0,786 | 0,312 | 0,512 | 0,409 | 0,225 | 0,573 |
| | 1 = 32 | 0,399 | 0,193 | 0,285 | 0,24 | 0,098 | 0,57 |
| | 1 = 64 | 0,234 | 0,106 | 0,172 | 0,15 | 0,047 | 0,265 |
| | 1 = 128 | 0,156 | 0,069 | 0,107 | 0,109 | 0,028 | 0,137 |
| Reindeer B lymph node | 1 = 2 | 3,162 | 3,5 | 3,26 | 2,603 | 2,724 | 3,202 |
| | 1 = 4 | 2,473 | 3,427 | 2,369 | 1,238 | 1,876 | 2,389 |
| | 1 = 8 | 1,398 | 2,263 | 1,521 | 0,945 | 1,035 | 2,368 |
| | 1 = 16 | 0,769 | 1,185 | 0,658 | 0,474 | 0,495 | 0,965 |
| | 1 = 32 | 0,323 | 0,703 | 0,355 | 0,243 | 0,271 | 0,263 |
| | 1 = 64 | 0,219 | 0,397 | 0,207 | 0,149 | 0,159 | 0,568 |
| | 1 = 128 | 0,12 | 0,21 | 0,105 | 0,095 | 0,099 | 0,248 |
| Cut-Off | | 0,175 | 0,149 | 0,149 | 0,149 | 0,228 | 0,148 |

The results of the analyses carried out in parallel at the NVI and the IRL on the set 2 are reported in Table 3. The analyses carried out on the five moose samples revealed that Herd-Chek BSE-Scrapie Ag test was able to detect all positive moose at all dilutions except for the moose E where positivity was revealed at dilutions of up to 1:4, indicating that this case was weaker than the others. TeSeE™ SAP Combi test was able to detect only two positive samples (moose C and D) thus showing a lower sensitivity compared to HerdChek BSE-Scrapie Ag test.

**Table 3. Results obtained from the set 2 of samples analyzed at NVI and IRL using three ELISA rapid tests and two western blot methods.** The O.D. represents the mean of duplicates /triplicates values obtained from each dilution sample by rapid tests. The values above the cut-off indicate positive sample and those below the cut-off indicate negative sample. Pos and Neg indicate positive and negative results by Western blot analyses.

| ID sample | Dilution | HerdCheckBSE/Scrapie Antigen Test with Bovine conjugate, short protocol | | TeSeE™ Sheep & Goat | TeSeE™ SAP | TeSeE™ Western Blot | SAF–Immunoblot | | | | |
|---|---|---|---|---|---|---|---|---|---|---|---|
| | | NVI–Optical density | IRL–Optical density | Optical density | Optical density | Sha31 | Sha31 | 6H4 | 9A2 | L42 | SAF84 |
| Moose A | undiluted | 3,382 | 3,317 | 0,176 | 0,017 | Pos | Pos | Pos | Pos | Pos | Pos |
| | 1 = 2 | 3,505 | 3,314 | 0,158 | 0,014 | Pos | Pos | Pos | Pos | Pos | Pos |
| | 1 = 4 | 3,5 | 3,307 | 0,072 | 0,015 | Pos | Pos | Pos | Neg | Neg | Pos |
| | 1 = 8 | 2,947 | 3,097 | 0,04 | 0,012 | Pos | Pos | Neg | Neg | Neg | Pos |
| | 1 = 16 | 2,164 | 2,6 | 0,047 | 0,013 | Pos | Pos | Neg | Neg | Neg | Pos |
| | 1 = 32 | 1,555 | 2,068 | 0,027 | 0,012 | Pos | Pos | Neg | Neg | Neg | Pos |
| | 1 = 64 | 1,011 | 1,039 | 0,021 | 0,012 | Pos | Pos | Neg | Neg | Neg | Pos |
| | 1 = 128 | 0,711 | 0,694 | 0,024 | 0,025 | Pos | Pos | Neg | Neg | Neg | Neg |
| Moose B | undiluted | 3,418 | 3,395 | 0,08 | 0,022 | Pos | Pos | Pos | Pos | Pos | Pos |
| | 1 = 2 | 3,478 | 3,374 | 0,101 | 0,012 | Pos | Pos | Pos | Pos | Pos | Pos |
| | 1 = 4 | 3,152 | 3,287 | 0,031 | 0,014 | Pos | Pos | Pos | Pos | Pos | Pos |
| | 1 = 8 | 2,28 | 2,878 | 0,025 | 0,014 | Pos | Pos | Pos | Pos | Pos | Pos |
| | 1 = 16 | 1,555 | 1,919 | 0,014 | 0,012 | Pos | Pos | Neg | Neg | Neg | Neg |
| | 1 = 32 | 0,815 | 1,527 | 0,027 | 0,01 | Pos | Pos | Neg | Neg | Neg | Neg |
| | 1 = 64 | 0,45 | 0,666 | 0,012 | 0,011 | Pos | Pos | Neg | Neg | Neg | Neg |
| | 1 = 128 | 0,277 | 0,354 | 0,013 | 0,012 | Pos | Pos | Neg | Neg | Neg | Neg |
| Moose C | undiluted | 3,372 | 3,24 | 0,027 | 0,461 | Pos | Pos | Pos | Pos | Pos | Pos |
| | 1 = 2 | 3,509 | 3,274 | 0,051 | 0,151 | Pos | Pos | Pos | Pos | Pos | Pos |
| | 1 = 4 | 3,5 | 3,263 | 0,061 | 0.024 | Pos | Pos | Pos | Pos | Pos | Pos |
| | 1 = 8 | 2,957 | 3,067 | 0,025 | 0,018 | Pos | Pos | Pos | Pos | Pos | Pos |
| | 1 = 16 | 2,291 | 2,571 | 0,032 | 0,013 | Pos | Pos | Pos | Pos | Pos | Pos |
| | 1 = 32 | 1,481 | 1,901 | 0,026 | 0,018 | Pos | Pos | Neg | Neg | Neg | Neg |
| | 1 = 64 | 0,994 | 1,292 | 0,019 | 0,036 | Pos | Pos | Neg | Neg | Neg | Neg |
| | 1 = 128 | 0,638 | 0,794 | 0,016 | 0,016 | Pos | Pos | Neg | Neg | Neg | Neg |
| Moose D | undiluted | 3,372 | 3,192 | 2.17 | 0,501 | Pos | Pos | Pos | Pos | Pos | Pos |
| | 1 = 2 | 3,5 | 3,198 | 1,987 | 0,446 | Pos | Pos | Pos | Pos | Pos | Pos |
| | 1 = 4 | 3,5 | 3,185 | 0,651 | 0,336 | Pos | Pos | Pos | Pos | Pos | Pos |
| | 1 = 8 | 3,432 | 3,408 | 1,624 | 0,202 | Pos | Pos | Pos | Pos | Pos | Pos |
| | 1 = 16 | 2,906 | 3,366 | 0.543 | 0,329 | Pos | Pos | Pos | Pos | Pos | Pos |
| | 1 = 32 | 2,139 | 2,878 | 0,672 | 0,129 | Pos | Pos | Pos | Pos | Pos | Pos |
| | 1 = 64 | 1,386 | 1,725 | 0,201 | 0,107 | Pos | Pos | Pos | Pos | Pos | Pos |
| | 1 = 128 | 0,949 | 1,076 | 0,118 | 0,053 | Pos | Pos | Pos | Pos | Neg | Neg |
| Moose E | undiluted | 0,627 | 0,332 | 0,02 | 0,012 | Pos | Pos | Neg | Pos | Neg | Pos |
| | 1 = 2 | 0,253 | 0,3 | 0,021 | 0,012 | Pos | Pos | Neg | Pos | Neg | Pos |
| | 1 = 4 | 0,137 | 0,196 | 0,013 | 0,011 | Pos | Pos | Neg | Neg | Neg | Neg |
| | 1 = 8 | 0,071 | 0,118 | 0,011 | 0,01 | Pos | Neg | Neg | Neg | Neg | Neg |
| | 1 = 16 | 0,044 | 0,079 | 0,011 | 0,011 | Neg | Neg | Neg | Neg | Neg | Neg |
| | 1 = 32 | 0,028 | 0,047 | 0,014 | 0,008 | Neg | Neg | Neg | Neg | Neg | Neg |
| | 1 = 64 | 0,029 | 0,042 | 0,017 | 0,009 | Neg | Neg | Neg | Neg | Neg | Neg |
| | 1 = 128 | 0,021 | 0,063 | 0,017 | 0,008 | Neg | Neg | Neg | Neg | Neg | Neg |
| Cut-Off | | 0,168 | 0,191 | 0,151 | 0,116 | ////// | ////// | ////// | ////// | ////// | ////// |

In addition, all tests detected easily PrP$^{Sc}$ in moose D which is less terminal-truncated than the other moose cases analysed.

On the basis of the results obtained from the comparison of the three rapid tests, it is therefore possible to state, also considering the concordance of the results obtained by the two different laboratories, that the HerdChek BSE-Scrapie Ag test is the most sensitive and robust diagnostic method.

### Confirmatory western blot

In order to assess the ability to confirm CWD cases identified by rapid screening tests in the European surveillance programme an evaluation of analytical sensitivity was also performed on TeSeE™ Western blot and SAF-Immunoblot. As reported in Table 3, both confirmatory western blot methods were able to detect the presence of PrP$^{Sc}$ in all dilutions of moose homogenate samples tested positive by the three rapid tests thus revealing a high analytical sensitivity, a fundamental requirement for a confirmatory diagnostic method. In particular, the best diagnostic confirmatory WB results were obtained using the mAb Sha31.

The Figs 2 and 3 (S1 and S2 Raw images) show the representative western blot analyses of two moose samples analysed by TeSeE™ Western blot and SAF-Immunoblot, respectively.

### Discussion

The emergence of CWD in Europe is, as with all prion diseases, a serious problem for both veterinary and public health. Based on the experience of the BSE crisis, several strategies have to be adopted by the European Commission for the management and control of the spread of these infectious diseases. Diagnostic surveillance programmes through the application of sensitive rapid tests, validated and authorised by the European Union have been a successful strategy to control BSE and Scrapie diseases in cattle and small ruminant in Member States.

The conclusions reported in the first EFSA Opinion on CWD mentioned that Norway detected the CWD cases with the TeSeE™ SAP Combi Kit and underlined the need of documenting the ability of other commercial screening tests to work equally well in detecting CWD in European cervids.

The results of the present study showed that the most used rapid diagnostic tests are also able to identify the different strains of CWD circulating in the Nordic countries, although with different analytical sensitivity. HerdChek BSE-Scrapie Ag test and HerdChek CWD Ag test resulted the most sensitive and robust, especially in the moose samples, while both western blot methods adopted in these diagnostic investigations were able to confirm the presence of PrP$^{Sc}$ in all samples positive by ELISA.

The level of analytical sensitivity shown by each of the three rapid tests is clearly related to the amount of PrP$^{Sc}$ present in the samples. In these investigations, we found that the lowest sensitivity was shown by all rapid tests in samples with lower levels of PrP$^{Sc}$ such as reindeer A and moose E, while no influence appears to be played either by cervid species nor the different strains.

The higher analytical sensitivity revealed by the HerdChek BSE-Scrapie and CWD Ag test compared to the TeSeE™ SAP Combi and TeSeE™ Sheep/Goat tests could also be due to their different processing method. The HerdChek Ag tests are based on their ability to bind only the pathological isoform of the prion protein and therefore it could be able to capture even intermediate isoforms of the PrP not yet completely folded into the β-sheet structure that could not resist the digestion with PK used in the other tests. On the other hand, the different analytical sensitivity could be due to the different detergents used for tissue preparation. The resistance of PrP$^{Sc}$ to the digestion action by PK has been shown to be strongly influenced by the type of

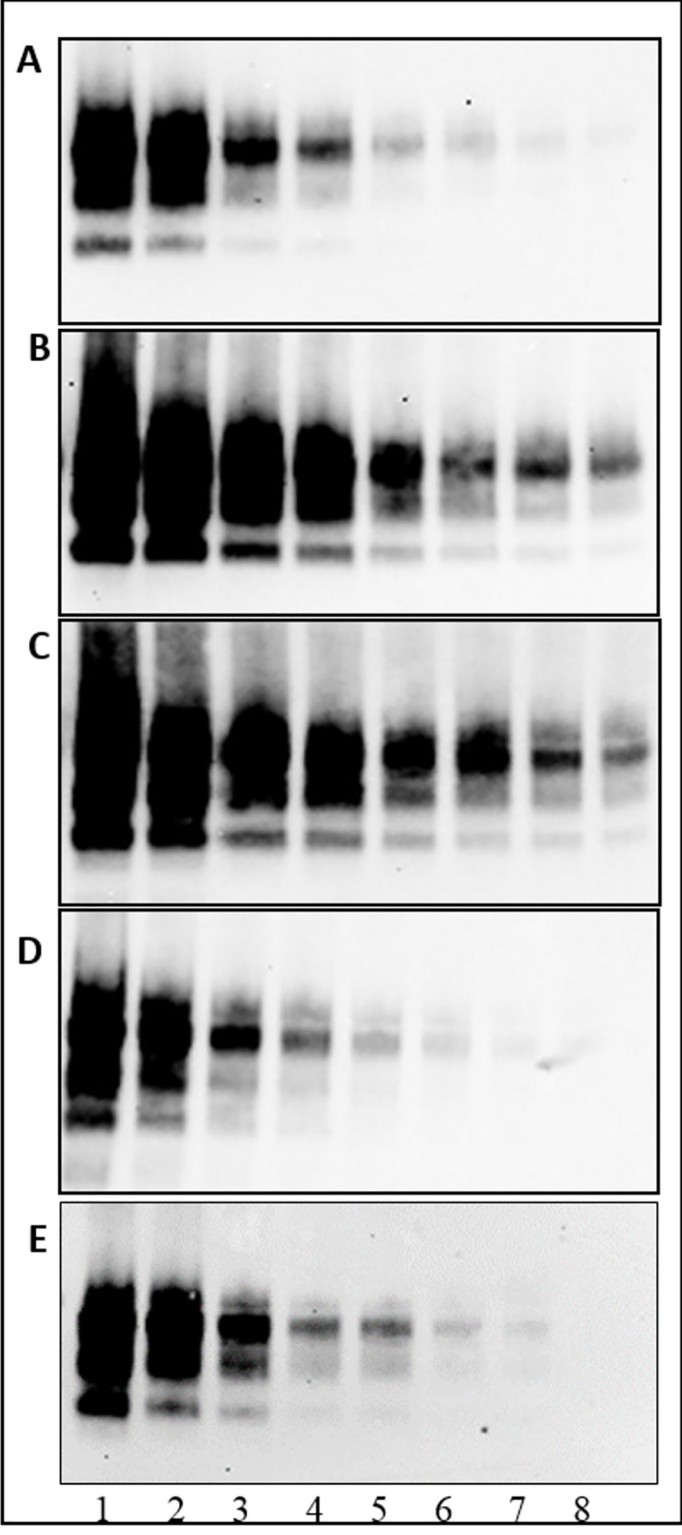

**Fig 2. Confirmatory SAF-immunoblot analysis of proteinase K-treated homogenates on dilution series from brain tissue of positive Moose D.** Lane 1 = undiluted; lane 2 = dilution 1:2; lane 3 = dilution 1:4; lane 4 = dilution 1:8; lane 5 = dilution 1:16; lane 6 = dilution 1:32; lane 7 = dilution 1:64; lane 8 = dilution 1:128. Membranes were probed with mAbs 6H4 (A), Sha31 (B), 9A2 (C), SAF84 (D), L42 (E).

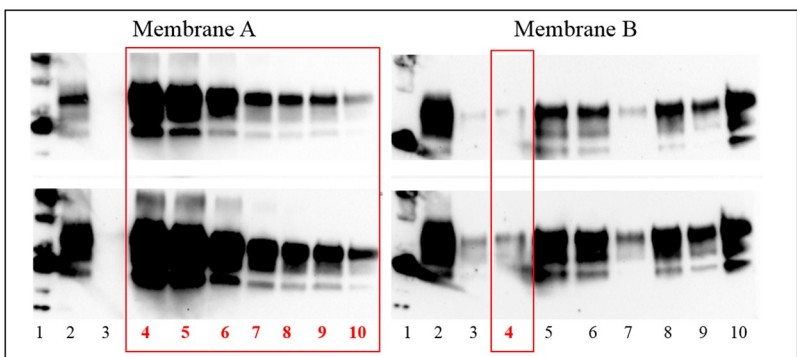

**Fig 3. TeSeE™ western blot analysis of proteinase K-treated homogenates on dilution series from brain tissue of positive Moose D (red rectangles) and positive Moose A, B, C.** Upper panel short exposition, lower panel longer exposition. Membrane A. Lane 1 = molecular weight; lane 2 = positive classical scrapie; 3 = negative moose control; lane 4 to 10: Moose D; lane 4 = undiluted; lane 5 = dilution 1:2; lane 6 = dilution 1:4; lane 7 = dilution 1:8; lane 8 = dilution 1:16; lane 9 = dilution 1:32; lane 10 = dilution 1:64. Membrane B. Lane 1 = molecular weight; lane 2 = positive classical scrapie; lane 3 = Moose E undiluted; lane 4 = Moose D dilution 1:128; lane 5 to 7: Moose A; lane 5 = undiluted; lane 6 = dilution 1:2; lane 7 = dilution 1:4; lane 8 to 9: Moose B; lane 8 = undiluted; lane 9 = dilution 1:2; lane 10 = Moose C undiluted.

detergents used [21]. An additional or alternative explanation to the differences in analytical sensitivity between the ELISA tests could be that one of the two antibodies used in the TeSeE ELISA tests performs poorly, especially the TeSeE SAP kit, of all the moose samples that lost the N-Terminal part [17], as far as the TeSeE ELISA tests work better with moose D and reindeer that are less N-terminal truncated.

The availability of a screening method with high analytical sensitivity is very important for detecting preclinical cases thus allowing a more effective control of the prion diseases but at the same time it can lead to the identification of false positive cases. Several cases in fact initially CWD reactive at the Norwegian laboratory with HerdChek BSE-Scrapie Ag tests were not then confirmed. Caution is therefore necessary to define a truly positive sample following a rapid screening test but only after the results obtained by official confirmatory methods such as Western blot and/or Immunohistochemistry.

Fortunately, our investigations revealed that the analytical sensitivity of both western blot methods applied in this study was similar or higher to that of rapid tests, validating their ability to confirm CWD cases identified by the screening tests. The sensitivity of a western blot method is greatly influenced by the choice of antibody. In our study the mAb that best identified the cases of Norwegian CWD was the core antibody Sha31, and the poorest antibodies were raised against the N-terminal of PrP, as the use of these later mAb might fail to detect PrP$^{Sc}$ in a positive case if its epitope is removed after digestion with PK as it was described for most of the Norwegian moose by Pirisinu et al. 2018 [17, 18]. Since these publications, two moose have been detected in Norway in which prions are not N-terminal truncated as previously described, and the present moose D is one of them. It is remarkable that the differences observed in analytical sensitivity between the ELISA tests are greatly reduced when analysing this moose where all tests performed well, as compared to the other N-Terminal truncated moose, indicating that the choice of antibodies is, at least partially, responsible for the difference in analytical sensitivity.

Limitations related to the small number of animal samples, due to the lack of tissue especially from reindeer, do not allow to draw exhaustive diagnostic conclusions. Nevertheless, the detection and confirmation at the NVI of the first CWD cases in reindeer, moose and red deer by TeSeE™ SAP Combi and TeSeE™ Western blot kits showed that these tests have good

diagnostic performances. We showed in this study that the approved ELISA test (HerdChek BSE-Scrapie Ag test) has even better sensitivity, and that both confirmatory WB protocols confirm all the positive results of the best ELISA results.

In conclusion, this study represents the first direct comparison between different diagnostic methods on European CWD cases. Despite the small number of samples, it is conceivable that the rapid and confirmatory diagnostic systems applied in Northern Europe for the CWD surveillance in cervid populations are reliable tools.

## Supporting information

**S1 Raw image. Cropped and uncropped image of SAF-Immunoblot.** Confirmatory SAF-Immunoblot analysis of proteinase K-treated homogenates on dilution series from brain tissue of positive Moose D and immunorevealed with five different anti-PrP monoclonal antibodies: 6H4 (A), Sha31 (B), 9A2 (C), SAF84 (D), L42 (E). Lane 1 = undiluted samples; lane 2 to 8 = two-fold dilutions of the sample from 1:2 to 1:128.
(TIF)

**S2 Raw image. Cropped and uncropped image of TeSeE™ western blot.** TeSeE™ Western blot analysis of proteinase K-treated homogenates on dilution series from brain tissue of four positive Moose A, B, C and D. Upper panel short exposition, lower panel longer exposition. Membrane A. Lane 1 = molecular weight; lane 2 = positive classical scrapie; 3 = negative moose control; lane 4 to 10: Moose D; lane 4 = undiluted; lane 5 to 10 = two-fold dilutions of the sample from 1:2 to 1:64. Membrane B. Lane 1 = molecular weight; 2 = positive classical scrapie; lane 3 = Moose E undiluted; lane 4 = Moose D dilution 1:128; lane 5 to 7: Moose A; lane 5 = undiluted; lane 6 = dilution 1:2; lane 7 = dilution 1:4; lane 8 to 9: Moose B; lane 8 = undiluted; lane 9 = dilution 1:2; lane 10 = Moose C undiluted.
(TIF)

## Author Contributions

**Conceptualization:** Maria Mazza, Linh Tran, Daniela Meloni, Elena Bozzetta, Sylvie L. Benestad.

**Data curation:** Maria Mazza, Elena Bozzetta, Sylvie L. Benestad.

**Formal analysis:** Maria Mazza, Daniela Meloni, Elena Bozzetta, Sylvie L. Benestad.

**Funding acquisition:** Maria Mazza, Jørn Våge, Sylvie L. Benestad.

**Investigation:** Linh Tran, Daniela Loprevite, Maria C. Cavarretta, Luana Dell'Atti.

**Methodology:** Maria Mazza, Linh Tran, Sylvie L. Benestad.

**Project administration:** Elena Bozzetta, Sylvie L. Benestad.

**Resources:** Daniela Loprevite, Maria C. Cavarretta, Jørn Våge, Knut Madslien.

**Writing – original draft:** Maria Mazza, Sylvie L. Benestad.

**Writing – review & editing:** Maria Mazza, Linh Tran, Jørn Våge, Knut Madslien, Tram T. Vuong, Elena Bozzetta, Sylvie L. Benestad.

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
