## [Decision Letter · Decision Letter 0]

2 Feb 2023

PONE-D-22-31903Are rapid tests and confirmatory western blot used for cattle and small ruminants TSEs reliable tools for the diagnosis of Chronic Wasting Disease in Europe?PLOS ONE

Dear Dr. Mazza,

Thank you for submitting your manuscript to PLOS ONE. After careful consideration, we feel that it has merit but does not fully meet PLOS ONE’s publication criteria as it currently stands. Therefore, we invite you to submit a revised version of the manuscript that addresses the points raised during the review process.

We look forward to receiving your revised manuscript.

Kind regards,

Human Rezaei

Academic Editor

PLOS ONE

Journal Requirements:

Reviewers' comments:

Reviewer's Responses to Questions

**Comments to the Author**

1. Is the manuscript technically sound, and do the data support the conclusions?

Reviewer #1: Yes

Reviewer #2: Yes

2. Has the statistical analysis been performed appropriately and rigorously? 

Reviewer #1: No

Reviewer #2: N/A

3. Have the authors made all data underlying the findings in their manuscript fully available?

Reviewer #1: Yes

Reviewer #2: Yes

4. Is the manuscript presented in an intelligible fashion and written in standard English?

Reviewer #1: Yes

Reviewer #2: Yes

5. Review Comments to the Author

Reviewer #1: The paper of Mazza et al., describes an evaluation approach of different prions commercially available diagnostic tests that are routinely used for cattle and small ruminant TSEs, applied in the present study to CWD prions circulating in the Scandinavian European area since 2016. The authors used different CWD isolates from two cervid species and tested them independently or in parallel in two different European laboratories. Indeed there is a need of having a CWD rapid diagnostic test to follow and detect CWD cases that appears in Europe given the strength of dissemination of this highly naturally contagious animal prions. Despite the very low number of CWD samples used in the submitted study, this short paper has the merit to bring interesting data on the first tries of the available prion diagnostic tests with CWD prions.

My First question is on the reasons why the authors have not tried all the CWD samples in parallel in the two different laboratories. This might have been more stringent for comparing the reproducibility and the efficiencies of the different diagnostic tests used, especially when different methodologies were used (ie; manual protocol (at IRV) versus robotic system (at NVI)for example).

- Line 62 : Prion accumulation in the brain leads to neurodegeneration and all the time to death. Why do the authors say « eventually to death » in this sentence?

-Lanes 93-96 : I Wonder why the Prionics WB test was not mentioned and eventually used in this comparative study.

-Lines 97-98 : What the authors mean by « are not comprehensive in the sentence starting with « Data on the performances… » ?

-Lines 119-120 : How many negative animals were used to prepare the negative brain pools of Moose and Reindeer ?

-Line 119 : Which type of lymph node was used ?

-Line 227-228 : The O.D. values of undiluted samples that are the only ones detected by the TSETMSAP Combi Kit are not reported on Table 2. Please clarify.

-Line 230 : As in Table 2 animals are designed by only their corresponding letter, please let this nomenclature in the text (Reindeer 17-CD2788 is Reindeer A).

-Lines 239-240 : I do not see on the western blot of Figure 3 any difference in PrPres molecular mass between Moose D and the other samples, as claimed by the authors. Please clarify.

-Why the WB blot of the Reindeer samples were not done and were not shown ?

-Lines 268 and 277 : In the discussion section, the authors talk about CWD strains. However, at the stage of the present analysis, it might be more appropriate to use the term isolate.

Minor points

-Homogenise Proteinase K as PK and not pK

-Homogenise g and not G or RCF when describing centrifugation steps.

-Lines 159 : The PK resistance of PrPSc is not only due to its beta-sheet structure but also to its aggregative state.

-Line 290 : replace than by that ?

-Figure 3 : is it necessary to show the two exposure times for WB revelation ? The longest one might be enough as it shows all the results.

Reviewer #2: In their manuscript entitled “Are rapid tests and confirmatory western blot used for cattle and small ruminants TSEs reliable tools for the diagnosis of Chronic Wasting Disease in Europe?”, Mazza and co-workers investigated the detection potential and limits of different commercial prion disease tests and tools to detect the prion agent responsible for Chronic Wasting Disease (CWD) in European cervids.

CWD appeared in 1967 in Colorado and is now spreading in North America and is out of control. Since 2016, CWD cases have been diagnosed in Norway, in Finland and Sweden. The European authorities decided to implement a surveillance program among cervids in order to design a control program of the disease expansion.

For this purpose, rapid and performant diagnosis tools are required and mandatory.

The present study investigates in the CWD context the performances of three commercially available tests, initially dedicated to diagnose TSE in small ruminants and cattle, and of two confirmatory western blot tests.

The experiments presented here are technically sound and the article is easy to read.

Here are my comments

1. The introduction part concerning the work done is a bit misleading. The authors state “performances of these tests were compared to two confirmatory western blot methods…”. However, Fig 1 does not show that set 1 has been tested using TeSeE western blot and confirmatory SAF-immunoblot but another test HerdCheck CWD Antigen Test has been used. This latter test is not presented in the materials and methods section. It has also not been tested on set 2. Could the authors comment this point? Have they performed the confirmatory tests they present on set 1?

2. Among the 41 CWD cases reported in Scandinavia, in reindeer, moose and red deer, 7 infected animals were chosen as positive controls. Could the authors explain their choice? Assessing the sensitivity of the tests initially designed for TSE detection in small ruminants and cattle, would require as many positive samples as possible.

3. Sensitivity is an important parameter in diagnosis tests, however specificity also. The authors never really mentioned this point in their manuscript. Could they make any comment and evaluate this parameter?

4. In their positive panel, reindeer and moose are present, but no red deer has been included. Since the rapid test sensitivity is not always the same for moose and reindeer (see results with TeSeE SAP Combi test), inclusion of red deer positive samples in the positive panel would have strengthened the presented results. Could the authors comment on this point or add some data regarding red deer?

5. How did the authors calculate their cut-off values? This is not explained in the manuscript?

6. Line 188, the authors mention that they used half of the Proteinase K concentration. Could the authors comment on this?

7. For a complete comprehension for the reader, some abbreviations would need to be explained (CC line 183, TMB line 184, RCF line 192, mAb line 198 (mAb is explained however later in the manuscript), ECL line 199, MP line 200, SAF line 248.

8. For homogeneity in their notation, the authors should choose between “g” and “RCF” in their materials and methods part when referring to centrifugation conditions, and also between “PK” (line 189 and “pK” (line 158) when referring to “proteinase K”.

9. The TBS composition is not mentioned (line 203).

10. Some concentrations are given as “microg ml” instead of “microg per ml” or “microg ml-1” (lines 205 and 216).

11. The transfer conditions onto PVDF membranes are not mentioned (line 209)

6. PLOS authors have the option to publish the peer review history of their article (what does this mean?). If published, this will include your full peer review and any attached files.

Reviewer #1: No

Reviewer #2: No

---

## [Author Response · Author response to Decision Letter 0]

20 Mar 2023

The answers to the Reviewers are reported in the file "Response to Reviewers".

---

## [Decision Letter · Decision Letter 1]

25 Apr 2023

PONE-D-22-31903R1Are rapid tests and confirmatory western blot used for cattle and small ruminants TSEs reliable tools for the diagnosis of Chronic Wasting Disease in Europe?PLOS ONE

Dear Dr. Mazza,

Thank you for submitting your manuscript to PLOS ONE. After careful consideration, we feel that it has merit but does not fully meet PLOS ONE’s publication criteria as it currently stands. Therefore, we invite you to submit a revised version of the manuscript that addresses the points raised during the review process.

We look forward to receiving your revised manuscript.

Kind regards,

Human Rezaei

Academic Editor

PLOS ONE

Journal Requirements:

Reviewers' comments:

Reviewer's Responses to Questions

**Comments to the Author**

1. If the authors have adequately addressed your comments raised in a previous round of review and you feel that this manuscript is now acceptable for publication, you may indicate that here to bypass the “Comments to the Author” section, enter your conflict of interest statement in the “Confidential to Editor” section, and submit your "Accept" recommendation.

Reviewer #1: (No Response)

Reviewer #2: All comments have been addressed

2. Is the manuscript technically sound, and do the data support the conclusions?

Reviewer #1: Yes

Reviewer #2: Yes

3. Has the statistical analysis been performed appropriately and rigorously? 

Reviewer #1: No

Reviewer #2: Yes

4. Have the authors made all data underlying the findings in their manuscript fully available?

Reviewer #1: Yes

Reviewer #2: Yes

5. Is the manuscript presented in an intelligible fashion and written in standard English?

Reviewer #1: Yes

Reviewer #2: Yes

6. Review Comments to the Author

Reviewer #1: The authors have brought clear answers and provided rational explanations to all the reviewer’s 1 queries. Given the importance of the data, I think that the figure provided for reviewer 1 showing the difference in electrophoretic migration between Moose C and moose D will be worth to be shown in the paper (and specify which WB method was used) as it will be of interest to the whole prion scientific community. However, it is to early to talk about terminal truncation of PrPSc of Moose C, since epitope mapping was not done in the present work. The term truncated should be sufficient at this stage.

Reviewer #2: The authors answered to all my concerns and made appropriate modifications in their revised version.

7. PLOS authors have the option to publish the peer review history of their article (what does this mean?). If published, this will include your full peer review and any attached files.

Reviewer #1: No

Reviewer #2: No

---

## [Author Response · Author response to Decision Letter 1]

11 May 2023

I now submit the final version of the manuscript.

---

## [Editor Report · Decision Letter 2]

12 May 2023

Are rapid tests and confirmatory western blot used for cattle and small ruminants TSEs reliable tools for the diagnosis of Chronic Wasting Disease in Europe?

PONE-D-22-31903R2

Dear Dr. Mazza,

We’re pleased to inform you that your manuscript has been judged scientifically suitable for publication and will be formally accepted for publication once it meets all outstanding technical requirements.

Kind regards,

Human Rezaei

Academic Editor

PLOS ONE
---

## [Editor Report · Acceptance letter]

21 Aug 2023

PONE-D-22-31903R2 

Are rapid tests and confirmatory western blot used for cattle and small ruminants TSEs reliable tools for the diagnosis of Chronic Wasting Disease in Europe? 

Dear Dr. Mazza:

I'm pleased to inform you that your manuscript has been deemed suitable for publication in PLOS ONE. Congratulations! Your manuscript is now with our production department. 

Kind regards, 

on behalf of

Dr. Human Rezaei 

Academic Editor

PLOS ONE